# Development and Validation of a New LC-MS/MS Method for Simultaneous Quantification of Ivacaftor, Tezacaftor and Elexacaftor Plasma Levels in Pediatric Cystic Fibrosis Patients

**DOI:** 10.3390/ph18071028

**Published:** 2025-07-10

**Authors:** Alessandro Mancini, Raffaele Simeoli, Luca Cristiani, Sara Cairoli, Fabiana Ciciriello, Alessandra Boni, Federico Alghisi, Chiara Rossi, Giacomo Antonetti, Carlo Dionisi Vici, Alessandro Giovanni Fiocchi, Renato Cutrera, Bianca Maria Goffredo

**Affiliations:** 1Division of Metabolic Diseases and Hepatology, Bambino Gesù Children’s Hospital, IRCCS, 00165 Rome, Italy; alessandro.mancini@opbg.net (A.M.); raffaele.simeoli@opbg.net (R.S.); sara.cairoli@opbg.net (S.C.); chiara.rossi@opbg.net (C.R.); giacomo.antonetti@opbg.net (G.A.); carlo.dionisivici@opbg.net (C.D.V.); 2Peadiatric Allergy and Cystic Fibrosis Research Unit, Bambino Gesù Children’s Hospital, IRCCS, 00165 Rome, Italy; luca.cristiani@opbg.net (L.C.); fabiana.ciciriello@opbg.net (F.C.); alessandra.boni@opbg.net (A.B.); federico.alghisi@opbg.net (F.A.); renato.cutrera@opbg.net (R.C.); 3Allergy Unit, Bambino Gesù Children’s Hospital, IRCCS, 00165 Rome, Italy; agiovanni.fiocchi@opbg.net

**Keywords:** cystic fibrosis, therapeutic drug monitoring (TDM), LC-MS/MS, CFTR modulators, pediatric patients

## Abstract

**Background**: “CFTR modulators” (also named “caftor”) have been developed and introduced into clinical practice to improve the functionality of defective CFTR protein. Therapeutic drug monitoring (TDM) is not currently used for CFTR modulators in routine clinical practice and there is still much to learn about the pharmacokinetic/pharmacodynamic (PK/PD) and the safety profiles of these drugs in a real-world setting. Moreover, therapeutic ranges are not yet available for both pediatric and adult cystic fibrosis (CF) patients. **Methods**: A new and sensitive liquid chromatography tandem mass spectrometry (LC-MS/MS) method for contemporary quantification of ivacaftor (IVA), tezacaftor (TEZ) and elexacaftor (ELX) in plasma samples has been developed and validated. The clinical performance of our method has been tested on samples collected during the routine clinical practice from n = 25 pediatric patients (aged between 7 and 17 years) affected by cystic fibrosis. This LC-MS/MS method has been validated according to ICH (International Council for Harmonisation of Technical Requirements for Pharmaceuticals for Human Use) guidelines for the validation of bioanalytical methods. **Results**: Our method fulfilled ICH guidelines in terms of accuracy, precision, selectivity, specificity and carry-over. Intra- and inter-day accuracy and precision were ≤15%. The 9-day autosampler stability was 90–100% for TEZ and ELX; meanwhile, it fell to 76% for IVA. An injection volume of 1 µL and a wider quantification range (0.1–20 µg/mL) represent a novelty of our method in terms of sensitivity and fields of application. Finally, the evaluation of PK exposure parameters for IVA revealed strong agreement with previously published reports and with results from the summary of product characteristics (SmPCs). **Conclusions**: This method could be adopted to contemporarily measure ELX/TEZ/IVA plasma levels for both PK studies and monitor therapy compliance, especially in case of poor or partial responses to treatment, or to evaluate drug–drug interactions when multiple concomitant medications are required. Considering also the high cost burden of these medications to the health system, a TDM-based approach could facilitate more cost-effective patient management.

## 1. Introduction

Cystic fibrosis (CF) is an autosomal recessive genetic disorder that affects 162,428 (144,606–186,620) people across 94 countries worldwide [1]. It is determined by variations in the gene sequence encoding for the CF transmembrane regulator (CFTR), a channel protein that acts as chloride and bicarbonate transporter [2]. To date, more than 2100 mutations have been identified (Cystic Fibrosis Mutation Database. Hospital for Sick Children, Toronto. http://www.genet.sickkids.on.ca/, accessed on 23 September 2024). However, the most frequent variant found in 85% of patients is F508del [3]. Although loss of function in CFTR protein leads to multisystemic dysfunction that involves the upper and lower airway, liver, pancreas, and gastrointestinal and reproductive tracts, lung disease is the most common cause of morbidity and mortality [4]. In fact, lung function is progressively impaired, being affected by chronic infections and multiple pulmonary exacerbations [5,6]. In this *scenario*, a pathological cascade including dehydrated airway surface liquid, reduced mucociliary clearance and dense secretions primarily lead to chronic pulmonary inflammation, irreversible lung architecture modification, respiratory failure and death [7]. Pharmacological therapies are mainly symptomatic and include mucolytics, airway hydrators, antibiotics and pancreatic enzymes [8,9,10,11]. However, in the last decade, “CFTR modulators” (also named “caftor”) have been developed and introduced into clinical practice, attempting to partially restore defective CFTR protein function [8,12]. In fact, these agents are able to counteract the intracellular dysfunction of CFTR protein, allowing significant improvement in terms of respiratory function, nutrition and quality of life for people with CF (pwCF) [8,12]. Actually, CFTR modulators can be distinguished into two groups defined as potentiators and correctors [13]. Potentiators (i.e., ivacaftor, IVA) increase anion transport by potentiating the channel-open frequency of CFTR protein at the cell surface [13]. Conversely, correctors (i.e., lumacaftor, LUM; tezacaftor, TEZ; elexacaftor, ELX) facilitate anion transport by correcting misfolding errors and promoting the CFTR protein migration to the cell surface [13]. Therefore, combinations of potentiators/correctors such as ivacaftor/lumacaftor, ivacaftor/tezacaftor and elexacaftor/tezacaftor/ivacaftor are often used in therapy to synergistically improve anion transport through F508del-CFTR protein via different mechanisms of action [13]. Recently, a novel oral drug formulation that combines ELX/TEZ/IVA (ETI) has been approved in both Europe (Kaftrio^®^) and the United States (Trikafta^®^) for treating CF patients aged 2 years and older carrying at least one F508del mutation [9,12,13,14,15]. Clinical trials for non-F508del patients are currently underway. The safety and efficacy of this triple combination have already been evaluated in several clinical studies showing a significant functional improvement and a positive outcome in patients bearing the F508del and other variants [13,16,17,18,19]. However, it is worth noting that high inter-individual variability in plasma levels of patients treated with “caftors” has been reported [9,20,21]. This variability is further enhanced by the drug–drug interactions (DDI) that involve CFTR modulators. In fact, these drugs are primarily metabolized in the liver through cytochrome P450 (CYP450) enzymes, mainly CYP3A4 and CYP3A5, which are inhibited by several drugs concomitantly administered to CF patients such as triazole antifungal agents, antivirals and macrolide antibiotics (i.e., clarithromycin and erythromycin) [12,22]. Moreover, physiological aspects that make children different from adults in terms of organ maturation and developmental changes represent an additional factor contributing to a higher inter-individual variability in pediatric patients compared to adults. Therefore, although therapeutic drug monitoring (TDM) is not currently indicated for CFTR modulators during routine clinical practice, there is an open debate on its clinical utility [4,8]. In fact, since their market authorization, different pharmacokinetic (PK) and pharmacodynamic (PD) studies have been conducted on both pediatric and adult CF patients [8,19,23,24,25,26]. However, more evidence is still needed to better elucidate the PK/PD behavior of these drugs in a real-world setting [4]. There is also an emerging adverse effect profile registered among CF patients, in particular concerning psychological effects such as anxiety, low mood, “brain fog” and insomnia, although it is not completely clear whether or not these effects could be ascribable to elexacaftor/tezacaftor/ivacaftor [4,27,28,29,30,31]. Therefore, monitoring plasma concentrations of CFTR modulators could be useful for dosing optimization in order to guarantee therapeutic drug exposures and limit adverse reactions [4,8]. Moreover, population PK (popPK) studies based on TDM results may represent a valid approach aimed at improving our knowledge on PK properties and individualized dose regimens in specific patient populations including pediatric subjects. To date, several bioanalytical methods based on liquid chromatography tandem mass spectrometry (LC-MS/MS) have been proposed for the quantification of different CFTR modulators in plasma and dried blood spot samples, although there are currently no commercially available assays [23,25,32,33,34,35,36,37,38,39,40]. However, examination of the existing literature often reveals complex chromatographic conditions and long sample preparation processes. In fact, an ideal analytical method should be robust, straightforward, reliable, and cost-effective.

Here, we have developed and validated a new sensitive LC-MS/MS method for simultaneous plasma-level determination of ivacaftor (IVA), tezacaftor (TEZ) and elexacaftor (ELX). Our method is characterized by an easy sample preparation and a wider calibration range (0.1–20 µg/mL), which is the same for each tested compound and allows the measurement of “caftors” in a low volume of plasma (50 µL). This method has been fully validated according to ICH guideline M10 on bioanalytical method validation and study sample analysis [41], and it has been applied to pediatric patients affected by cystic fibrosis under steady-state treatment with Kaftrio^®^. The aim of our study is to propose a viable assay for the implementation of “caftor” therapeutic drug monitoring in the bioanalytical laboratory during routine clinical practice. In fact, TDM for CFTR modulators could be useful not only to improve our knowledge about the PK/PD of these drugs in special populations but also to allow individualized dosing strategies and provide more cost-effective management, particularly considering the high cost burden of these medications to the health system.

## 2. Results

### 2.1. Calibration Curve and Linearity Evaluation

Linear calibration curves were used to cover the range 0.1–20 µg/mL for IVA (y = 0.138983 *** x + 0.021339, R^^^2 = 0.996), TEZ (y = 0.129561 * x + 0.005296, R^^^2 = 0.998) and ELX (y = 0.050441 * x + 0.004105, R^^^2 = 0.998) (Appendix A).

Linearity was evaluated on five calibration curves realized on five different days. Each calibrator was quantified within 15% of the target concentration. Moreover, to further evaluate the linearity of each calibration curve, back-calculated concentrations for ivacaftor, tezacaftor and elexacaftor calibration standards were evaluated, and accuracy (expressed as % bias) was also computed by comparing calculated to nominal concentrations. For each calibration standard, the % bias was within the acceptable value of ≤15%.

Additionally, in order to assess the validity of linear regressions, analysis of residuals for each calibration function was performed showing a Max % Residual of −7.6, −13.6 and −4.7 for IVA, TEZ and ELX, respectively.

### 2.2. Selectivity and Specificity

Drug-free plasma samples were analyzed to evaluate possible endogenous interferences with “caftor” detection. As reported in Figure 1A–C, STD 6 samples spiked with IS did not show interfering peaks within ivacaftor, tezacaftor and elexacaftor chromatogram. The median signal of these blank samples was below 20% of the LLOQ, thereby ensuring the selectivity of the method.

LLOQ concentration was 0.1 µg/mL and was determined by dissolving decreasing concentrations of ivacaftor, tezacaftor and elexacaftor powder in pooled drug-free plasma (Figure 2A–C). LLOQ was identified and confirmed with an accuracy and precision within 20%. Moreover, intra- and inter-assay accuracy and precision at the LLOQ level were measured from the six-point calibration curve. Results are reported in Table 1 and Table 2.

The validity of our analytical method was further confirmed by evaluating system suitability in accordance with United States Pharmacopeia (USP) [42]. In particular, we have used Agilent MassHunter Qualitative Analysis software (version 10.0, Agilent Technologies) to calculate the following parameters: Resolution (Rs), Capacity Factor (k′), Peak Symmetry (As) and Tailing Factor (T). As reported in Appendix A, all these values were within the acceptable ranges as previously described [43,44].

### 2.3. Accuracy and Precision

Intra- and inter-assay accuracy and precision were evaluated for LLOQ, L-QC, M-QC and H-QC (Table 1 and Table 2). Both parameters agreed with ICH guideline M10 on bioanalytical method validation and study sample analysis. Specifically, the intra- and inter-assay accuracy (reported as mean %bias) was ≤15% at each QC level. Similarly, precision (expressed as %CV) was ≤15% for both intra- and inter-assays at the low, medium and high QC levels.

In order to assess the presence of carry-over, IS-spiked blank samples were run in triplicate, following the highest calibration point. According to ICH guideline M10 on bioanalytical method validation, the median signal of these blank samples was less than 20% of the LLOQ and 5% of the IS, confirming the absence of carry-over.

Analyses performed to evaluate the matrix effect and IS-normalized matrix effect resulted in values within the acceptable range (85–115%) for ivacaftor, tezacaftor and elexacaftor.

Similarly, the extraction recovery (ER%) ranged from 90 to 95% (CV% < 15%). Results are reported in Table 3.

### 2.4. Evaluation of Stability

Both short- and long-term stability were evaluated on prepared QC samples kept at room temperature for a maximum of 9 days. After 24 h from QC sample preparation and first assessment (Time 0), the stability was around 100% for all QC samples. After 9 days from day 0, the stability decreased exclusively for L-, M- and H-QC levels of ivacaftor (73.54–78.27%). Results are reported in Table 4.

We have also evaluated the stability of prepared low-, medium- and high-QC samples following one cycle of freezing (−80 °C) and thawing (after 12 days). The calculated stability for ivacaftor, tezacaftor and elexacaftor was 100–107% after 12 days. Patient samples (n = 3 T0 and T4) were randomly selected and re-analyzed after being stored at −80 °C over a period of five months. Compared to the first measurement (time 0), the stability ranged from 85.48 ± 19.83% (mean ± SD) for IVA T4 to 90.55 ± 2.42% for ELX T0 (Appendix A).

### 2.5. Robustness Assessment

The robustness of the proposed method was evaluated by applying deliberate minor changes in the experimental conditions as follows: sample preparation and analytical run submission were performed by different operators in order to test the effects of inter-operators variability among various analytical sessions; two analytical runs have been carried out by changing room temperature (25 ± 2 °C) to mimic changes that could partially affect instrument working conditions; mobile phases have been prepared by using reagents from different batches (including acetonitrile, used for both mobile phase B preparation and protein precipitation); we performed two analytical runs by changing the pH of mobile phase A (2.70 ± 0.3) in order to evaluate the effect of pH variations on the measured drug concentrations.

The results of these robustness tests indicated that our method remained reliable and produced consistent results despite the minor changes that may occur during the experimental procedures.

### 2.6. Evaluation of Method Greenness, Blueness and Whiteness

For the greenness assessment of our method, we used the AGREE approach. This includes all the GAC principles and provides an overall score, which considers the weight of each criterion [45]. In our study, whilst the default weight of 2 was applied to the majority of criteria, we applied a weight of 3 to criteria 2, 6 and 7. In fact, in our opinion, the use of small sample volumes is an important aspect to consider in a green analytical technique since it allows the use of lower volumes of solvents and, as a consequence, a reduction in the amount of waste. Similarly, the absence of derivatization steps in our method reduces the risk of exposure to toxic and polluting reagents for the operators. Finally, as a result of criterion 2, we also applied a weight of 3 to criterion 7, considering that a reduced amount of waste is another important aspect to consider for laboratories performing both research and diagnostic analyses. The final score calculated for our method was 0.60, and a pictogram is depicted in Figure 3A showing both environmentally friendly aspects and hazardous characteristics. In particular, the small sample volume required and the absence of derivatization steps represented the greenest hallmarks of our method. Conversely, criteria 3, 9 and 11 highlighted the most hazardous subsections, corresponding to off-line sampling, energy consumption by the LC-MS/MS instrument and the use of toxic solvents (acetonitrile), respectively.

In terms of the method’s practicality, the pictogram reported in Figure 3B represents the Blue Applicability Grade Index (BAGI), a new GAC metric that combines both the practicability and greenness of analytical methods [46]. The different hues of blue indicate different degrees of method applicability, where dark blue denotes high compliance, blue represents moderate compliance, light blue indicates low compliance, and white signifies non-compliance. In our method, the strongest practicability points were represented by the absence of preconcentration steps during sample preparation, the use of commercially available reagents and the low volume of sample required (<100 µL). Conversely, the use of instrumentation not commonly available in most laboratories (i.e., LC-MS/MS) corresponded to the weakest point in terms of applicability.

Finally, the whiteness of our bioanalytical method was also evaluated. The idea of white analytical chemistry (WAC) represents a possible compromise between greenness and functionality [47]. Here, we have used RGBfast, an improved version of the RGB model, since it also includes the ChlorTox Scale as one of the evaluation criteria [48]. Figure 3C shows a calculated RGBfast score of 52, indicating that our method, despite its limited greenness, was endowed with a higher whiteness evaluation compared to other methods [35].

### 2.7. Measurement of “Caftor” Plasma Levels in Patients Affected by Cystic Fibrosis

The clinical performance of our validated method was tested on n = 25 samples from pediatric patients affected by cystic fibrosis. Table 5 reports plasma concentrations of IVA, TEZ and ELX measured before (T0) and 4 h after (T4) the morning dose assumption of Kaftrio^®^ (corresponding to IVA T_max_). For IVA, the plasma concentration at 12 h (T12) was obtained by using C4h*e^−β(t12−t4)^, and the AUC_0–12h_ was calculated to assess the systemic exposure within the interval between two doses (Table 5). Thereafter, AUC_0–12h_ values were multiplied by 2 in order to perform a regression analysis of T0 concentrations *versus* AUC_0–24h._

A linear correlation was observed between IVA T0 (C_trough_) and the corresponding AUC_0–24h_ values, with a Spearman r correlation coefficient of 0.94 (95% confidence interval, 0.86–0.97) (Figure 4A). Analysis of residuals showed a % variation coefficient (%CV) between measured and predicted AUC_0–24h_ values of 0.02 ± 13.12 (mean ± SD) (Figure 4B).

The derived regression equation (y = 27.6 *-x + 4.30) was used to back-calculate the C_trough_ (T0) concentrations that result in a median (IQR) value of 0.80 (0.42–1.32) µg/mL.

## 3. Discussion

To date, different pharmacokinetic (PK) and pharmacodynamic (PD) studies have been conducted on both pediatric and adult CF patients [8,19,23,24,25,26]. However, what constitutes the therapeutic range for the CFTR modulators still remains an open question [8]. Actually, our knowledge is limited and sometimes derives from the SmPCs or from sparse real-life studies in which data from both children and adults are included [35]. These aspects highlight the necessity of obtaining more evidence in order to better elucidate the PK/PD behavior of these drugs in a real-world setting [4]. Another important aspect that underlays the utility of monitoring “caftor” plasma concentrations is represented by the adverse reactions that involve also psychological effects [4,27,28,29,30,31]. However, in two previously published reports, a reduction in ETI dose led to improved mental health [28,49]. Finally, although the use of “caftors” in pregnancy is “off-label”, emerging evidence suggests that the continuation of therapy with CFTR modulators is able to prevent a clinical decline during pregnancy [50,51]. Considering the altered PK profile that characterizes this critical time, further studies aimed at evaluating the PK/PD parameters of ivacaftor/tezacaftor/elexacaftor in pregnancy and breastfeeding are needed [52].

Based on these considerations, TDM of CFTR modulators could be useful for improving dosing optimization in different clinical scenarios, ensuring therapeutic drug exposures and limiting adverse reactions [4,8]. Recently, a retrospective analysis conducted on adult patients with cystic fibrosis, who were assuming ETI combination, has highlighted the value of utilizing a TDM program, underlying the need for further PK studies [53].

Our aim was to develop and validate a new LC-MS/MS method for simultaneous plasma-level determination of ivacaftor (IVA), tezacaftor (TEZ) and elexacaftor (ELX). The method presented here has been fully validated following ICH guideline M10 on bioanalytical method validation and study sample analysis [41], and it has been applied to pediatric patients affected by cystic fibrosis under steady-state treatment with Kaftrio^®^.

Currently, there are no commercially available assays for “caftor” determination in biological matrices. Therefore, several bioanalytical methods based on liquid chromatography tandem mass spectrometry (LC-MS/MS) have been proposed to detect CFTR modulators in plasma samples [23,25,32,33,34,35,36,37,40]. In Appendix A, we report the analytical characteristics of our method compared to those already published. Although the run time and the use of deuterated internal standard (IS) do not represent a significant difference compared to other published reports, a wider calibration range (0.1–20 µg/mL), which is the same for each tested compound, allows the measurement of “caftor” plasma levels in a larger concentration window. In fact, as reported in Table 5, the median (IQR) level of elexacaftor 4 h after daily morning intake of Kaftrio^®^ was 17.85 (13.27–24.10) µg/mL. Thanks to the wide calibration range, we were able to measure ELX concentrations without diluting samples. This aspect could be particularly useful for studies aimed at exploring the PK of CFTR modulators in both children and adults for whom the high-interindividual variability could be responsible for significant oscillations in drug plasma levels.

Sensitivity can be considered as another important advantage of our method since we are able to quantify drug concentrations at the LLOQ level by using a very small injection volume (1 µL). Moreover, low injection volumes in mass spectrometry offer several advantages, primarily related to improved chromatographic resolution, enhanced sensitivity, and reduced sample consumption. In fact, by reducing the volume of sample injected into the separation column, band broadening is reduced, leading to sharper, better-defined peaks and improved resolution of closely eluting compounds. This aspect can be particularly beneficial in complex samples where many compounds show narrow retention times. Furthermore, lower injection volumes can lead to higher sensitivity, by increasing the concentration of analytes reaching the detector, especially when micro- or nano-flow chromatography techniques are used [54].

Finally, to further validate our bioanalytical method, we performed system suitability assessment according to the United States Pharmacopeia (USP) [42]. In detail, the Resolution (Rs), Capacity Factor (k’), Peak Symmetry (As) and Tailing Factor (T) resulted within the acceptable ranges [43,44].

In our method, sample preparation is simple and does not require multiple steps (i.e., LLE, SPE and/or derivatization). Moreover, it is carried out from 50 µL of plasma. This low sample volume is favorably suitable to both adults and pediatric patients with CF. In fact, for studies involving pediatric patients, ethical and physiological concerns do not allow the collection of large volumes of blood through multiple repeated tests. Additionally, the use of small sample volumes and the absence of derivatization steps are two important aspects in the overall greenness evaluation and represent the greenest hallmarks of our method. Conversely, off-line sampling and the energy consumption by the LC-MS/MS instrument are the most hazardous subsections in our AGREE score. However, considering that blood samples from both hospitalized and outpatients cannot be collected in the TDM laboratory, a different sampling procedure could not be performed. Similarly, the analytical technique used in our study (i.e., LC-MS/MS) still represents the gold standard for TDM analysis, and alternative immunometric methods based on immunoenzymatic reactions are not advisable due to their low sensitivity and specificity. However, the overall AGREE score calculated for our method (0.60) can be classified as “acceptable green”, and it is in line with previously published LC-MS/MS methods [55], including one report aimed to simultaneously quantify ivacaftor, tezacaftor, and elexacaftor in plasma samples of CF patients [35].

A higher score has been calculated when evaluating the blueness of our developed method. To this aim, we have used the Blue Applicability Grade Index (BAGI), a metric that incorporates 10 criteria to assess both the practicability and greenness of different assays [46]. In fact, this tool is used for evaluating the practicability of a method in analytical chemistry and the attributable score ranges from 25 to 100 (the higher the score, the more practical the method). For a method to be considered “practical,” the final rating score should exceed 60. Here, we have calculated a BAGI score of 75, thus denoting the good practicability of our method.

Finally, the white analytical chemistry (WAC) expands the concept of green analytical chemistry (GAC) and highlights the importance of achieving the best possible compromise between greenness and functionality [47]. In fact, WAC refers to the Red–Green–Blue (RGB) model of a light color, in which greenness is one of the three primary components of whiteness [56]. Red is assigned to analytical performances such as the accuracy, precision and sensitivity, whereas blue is assigned to the practical and economic aspects [46]. The overall quality of the method is expressed by whiteness. Here, we have used an improved version of the RGB model, called RGBfast, which also includes the ChlorTox Scale, a recently developed greenness indicator [48,57]. As depicted in Figure 3C, our method seems “whiter” compared to a similar published report [35], despite limited greenness. However, in accordance with a previously provided definition, a whiter method is one that is more comprehensive and better overall, but at the same time not necessarily greener. In fact, certain methods may be highly green but not very functional in the red or blue areas, and so would not represent the desired compromise [48].

The clinical applicability of our validated method was tested by measuring ETI plasma concentrations in pediatric patients followed at our center during routine clinical practice. In particular, blood sampling was performed before (C_trough_, T0) and 4 h after (T4) the morning intake of Kaftrio^®^. Our data for IVA revealed a median (IQR) C_trough_ value of 0.79 (0.37–1.34) µg/mL, which agreed with the drug registration documents (Trikafta^®^ mean ± standard deviation, SD = 0.63 ± 0.26 µg/mL) [18] and with previously published studies on both children (mean ± SD= 0.71 ± 0.20 µg/mL) and adult patients (mean ± SD = 0.90 ± 0.37 µg/mL) [35]. For both TEZ and ELX, we found some discrepancies between our results and other reports including the medication SmPC (Kaftrio^®^) [22]. For example, our results for TEZ C_trough_ (median, IQR = 1.37, 0.72–1.87 µg/mL) confirm the minimum concentration (C_min_) values shown in a study by Vonk S.E.M. and colleagues (median, range = 1.3, 0.5–2.8 µg/mL) (2024) [58], but not those described in the registration documents (Trikafta^®^ mean ± SD = 2.10 ± 0.82 µg/mL) [18]. Conversely, C_trough_ values measured in our study for ELX (median, IQR = 8.73, 5.82–12.94 µg/mL) were higher than those reported in both SmPCs (Trikafta^®^ mean ± SD = 4.05 ± 2.07 µg/mL) [18] and previous studies (median, range = 2.8, 0.9–7.9 µg/mL) [58]. However, a recent study conducted in a real-world cohort of adults with cystic fibrosis has reported an AUC_0–24h_ for elexacaftor and tezacaftor ranging from 58.7 to 422.9 mg⋅h/L and 38.0 to 207.7 mg⋅h/L, respectively [26].

Reasons behind these discrepancies may be attributed to differences in the nature of these studies. In fact, our method has been applied to samples collected from pediatric patients (age range: 7–17 years) in a real-life setting, which significantly differs from the fixed conditions established during the registration studies. In particular, plasma samples tested with our validated method were obtained from CF patients who showed clinical stability or were experiencing respiratory exacerbations during the follow-up visits in a clinical practice scenario. Therefore, the inter-individual variability together with the underlying health conditions could be important factors affecting the PK response to these drugs.

In this context, it is also important to consider possible drug–drug interactions as a consequence of multiple concomitant medications. In fact, the cytochrome P450 3A (CYP3A) system is the main metabolizing route for all three drugs. Many drugs commonly used in CF patients act as inhibitors or inducers of both CYP3A4 and CYP3A5 isoforms (for example, rifampicin, azole antifungals or macrolides antibiotics) leading to DDIs that could potentially alter “caftor” plasma concentrations [8]. Similarly, the bioavailability of CFTR modulators is significantly affected by the concomitant ingestion of food containing a high percentage of fat [8]. In fact, it has been reported that the AUC of elexacaftor is increased by 1.9- to 2.5-fold when assumed with a fatty meal, whereas exposure to TEZ is not affected by the simultaneous ingestion of fatty food (Kaftrio^®^) [22]. Therefore, a real-world setting could be characterized by different sources of variabilities that in a randomized clinical trial (RCT) are significantly reduced. Patients tested in this study were concomitantly receiving therapies with pancreatic enzymes, bronchodilators, antibiotics (including rifampin, carbapenems, macrolides, beta-lactams, lipopeptides, fluoroquinolones) and proton pump inhibitors. Among these medications, the Kaftrio^®^ dosing regimen should be adjusted when clarithromycin (macrolide antibiotic) and/or rifampin are co-administered [22]. Therefore, this aspect underlies not only the utility of monitoring ETI concentrations in a real-life setting but also the different sources of variability that make routine clinical practice different from RCTs.

Additionally, children are physiologically different from adults in terms of organ maturation and developmental changes. This aspect adds a further source of inter-individual variability that could partially explain the differences observed in our C_trough_ values compared to the studies involving both adults and pediatric subjects [35,58]. However, it is also worth noting that within the pediatric population, changes in body weight (< or ≥30 Kg) require a different dosage (Kaftrio^®^ SmPC) [22]. Therefore, an age range of 7–17 years is not enough to guarantee dosage homogeneity, introducing an additional variable that could affect plasma concentrations. Perhaps, it is possible that tezacaftor and elexacaftor could be more susceptible to these “real-life” sources of variability, thus explaining the observed discrepancies.

In order to better elucidate the PK properties of ivacaftor in a real-life pediatric setting, an additional blood sampling at 4 h (T4) following Kaftrio^®^ intake (considered as the time to reach the maximum plasma concentration, T_max_, for IVA) was included in our TDM protocol. After estimating plasma concentrations of ivacaftor at 12 h, an AUC_0–12h_ was calculated for each patient. As reported in Table 5, our results for IVA C_max_ (median, IQR = 1.49, 1.07–2.43) and AUC_0–12h_ (median, IQR = 11.43, 7.93–19.45 μg/mL*h) agreed with the drug registration documents (C_max_, mean ± SD =1.24 ± 0.34 µg/mL and AUC_0–12h_, mean ± SD =11.7 ± 4.01 μg/mL*h, Kaftrio^®^) [22] and with previously published studies on both children and adult patients [23,24,26,33,53,59,60].

Similarly, a significant correlation (*p* < 0.001) was found between C_trough_ (T0) and AUC_0–24h_ values for ivacaftor (Figure 4A), confirming previous published evidence [58]. This strong correlation was further confirmed by the analysis of residuals that showed a low %CV between measured and predicted AUC_0–24h_ values of 0.02 ± 13.12 (mean ± SD) (Figure 4B). Moreover, by using the derived regression equation, we were able to back-calculate the C_trough_ values for IVA. The median (IQR) value for the calculated C_trough_ was 0.80 (0.42–1.32) µg/mL. This median value established with the regression equation (Figure 4A) presented high correspondence with our observed C_trough_ (median, IQR = 0.79, 0.37–1.34 µg/mL) and with the C_min_ ranges found in the registration documents (Trikafta^®^ mean ± SD = 0.63 ± 0.26 µg/mL) [18]. These results agree with those reported by Vonk S.E.M and colleagues (2024) on the back-calculated C_trough_ values for IVA (mean ± 1.96 SD = 0.7 ± 0.1–1.2 µg/mL), and suggest that C_trough_ could be a good predictor of the systemic exposure to ivacaftor [58]. In fact, although the AUC represents the gold standard for determining drug exposure, it requires multiple blood sampling during one dosing interval. As a consequence, this practice is invasive and poorly feasible in both clinical studies and during the routine clinical practice, especially when neonates and children are involved. Therefore, measuring only one C_trough_ sample instead of performing multiple blood sampling could represent an important advantage in evaluating the systemic exposure to CFTR modulators. In our study, we were not able to assess the correlation between C_trough_ and AUC_0–24h_ also for tezacaftor and elexacaftor. In fact, based on a T_max_ of 3 h for TEZ and 6 h for ELX [8], we were not able to calculate the AUC_0–24h_. However, Vonk S.E.M. and colleagues (2024) reported a significant correlation between C_min_ and the AUC_0–24h_ values for tezacaftor and elexacaftor [58]. Therefore, considering the agreement between our data for IVA and those previously reported [58], a similar result should also be predictable in our pediatric setting for TEZ and ELX.

Both short- and long-term stability were evaluated on prepared QC samples kept at room temperature in the autosampler for a maximum of 9 days. After 24 h from QC sample preparation and first assessment (Time 0), stability was around 100% for all QC samples. After 9 days from day 0, stability decreased exclusively for the L-, M- and H-QC of ivacaftor (73.54–78.27%), probably due to different chemical properties of ivacaftor compared to tezacaftor and elexacaftor. However, a similar result has been already observed in a recently published method in which the authors evaluated the stability of compounds in EDTA plasma after four freeze–thaw cycles, and for ten days in the autosampler after sample preparation. Following ten days in the autosampler, the observed remaining % for the H-QC of ivacaftor was 88%, compared to the H-QC of tezacaftor and elexacaftor (both 99.8%) [40].

Finally, n = 3 T0 and T4 patients’ samples were re-tested after a storage at −80 °C for a period of five months. ELX/TEZA/IVA concentrations showed a stability higher than 85% of the first measurement, meeting the criteria of the remaining percentages being between 85% and 115%. Therefore, in order to overcome the absence of mass spectrometry facilities, centers involved in PK studies could collect and store samples at −80 °C before sending them to TDM laboratories for the evaluation of ELX/TEZA/IVA plasma levels.

One limitation of this study may be the absence in our LC-MS/MS method of active metabolite determination. In fact, elexacaftor, ivacaftor and tezacaftor are metabolized through CYP3A isoforms, and the concomitant assumption of potent CYP3A inhibitors or inducers would lead to changes in active metabolite concentrations requiring an adjustment of caftor dosage [37]. However, considering that both elexacaftor M23-445 and M1-TEZ have similar potencies to their parent compounds, whereas M1-IVA has shown a 1/6 potency of IVA [8], our aim was to primarily focus on the measurement of ELX/TEZA/IVA plasma concentrations. Moreover, due to the lack of therapeutic ranges for the active metabolites, future studies will be required to assess the utility of TDM for these compounds.

Similarly, the clinical applicability of this method has been tested on plasma samples of pediatric patients aged from 7 to 17 years. Considering the recent approval of ETI combination for treating CF patients aged 2 years and older, a further limitation of our study could be represented by the lack of PK data collected from younger children in a real-world clinical setting. Therefore, future reports will be needed in the next future to fill this knowledge gap.

## 4. Materials and Methods

### 4.1. Powders and Reagents

Ivacaftor (IVA), Ivacaftor-D9 (IVA-D9), Tezacaftor (TEZ), Tezacaftor-D4 (TEZ-D4), Elexacaftor (ELX) and Elexacaftor-D3 (ELX-D3) were obtained from Toronto Research Chemicals (North York, ON, Canada). LC-MS/MS acetonitrile (ACN), 2-propanol and formic acid were purchased form Biosolve Chemicals (Dieuze, 57260, France). Water for LC-MS/MS applications was provided by VWR International (Radnor, PA, USA). LC-MS/MS methanol was obtained from Sigma Aldrich Srl (Milan, Italy).

### 4.2. Preparation of Stock and Working Solutions

IVA, IVA-D9, TEZ, TEZ-D4, ELX and ELX-D3 stock solution were dissolved in LC-MS/MS methanol to obtain stock solutions at concentrations of 1 mg/mL. The stock solutions were all stored at −80 °C until use. A working mixed solution of IVA, TEZ and ELX was prepared for the sample preparation by diluting the stock solutions of IVA, TEZ and ELX 1:10 in LC-MS methanol. The same was applied to the IS working mixed solution including IVA-d9, TEZ-d4 and ELX-d3.

### 4.3. Calibrators and Quality Control (QC) Samples

A calibration curve was prepared with six points (excluding blank samples) by performing serial dilutions from the working solution of IVA, TEZ and ELX stock solution (1 mg/mL) in drug-free plasma. The calibrator (CAL) concentrations were as follows: 0.1, 1, 5, 10, 15 and 20 µg/mL. Samples resulting in a concentration above the highest calibration (ULOQ) point were diluted using pooled blank plasma and re-analyzed.

Quality control (QC) samples for accuracy and precision runs were prepared at 4 concentration levels within the calibration curve range: lower limit of quantification (LLOQ) at 0.1 µg/mL and n = 3 additional QCs at 0.3 µg/mL (low QC), 8.0 µg/mL (medium QC) and 18 µg/mL (high QC). LLOQ was assessed by dissolving decreasing concentrations of IVA, TEZ and ELX powders in drug-free plasma. Subsequently, n = 40 replicates were analyzed in five different analytical sessions. LLOQ showed accuracy and precision within 20%.

### 4.4. Human Samples

Drug-free plasma was obtained from healthy donors, who gave their informed consent at the Blood Transfusion Centre of the Bambino Gesù Children’s Hospital (Rome, Italy), and was used to prepare LLOQ, calibration standards (CAL), low-, medium- and high-quality controls (QCs), and blank samples for the evaluation of the method’s selectivity and specificity. Pooled drug-free plasma samples were frozen at −20 °C until use.

The clinical applicability of our validated method was tested on n = 25 plasma samples from pediatric patients with a median (range) age of 9.5 (7–17) years and affected by cystic fibrosis (CF). These patients (n = 14 males and n = 11 females) were followed-up during the routine clinical practice at Pneumology and Cystic Fibrosis Unit of Bambino Gesù Children’s Hospital, and were concomitantly receiving therapies with pancreatic enzymes, bronchodilators, antibiotics (including carbapenems, macrolides, beta-lactams, lipopeptides, fluoroquinolones) and proton pump inhibitors.

In order to evaluate systemic exposure to each CFTR modulator, drug levels were measured at steady-state by collecting EDTA–whole blood samples before (T0, corresponding to the minimum plasma concentration, C_trough_,) and 4 h after the morning dose intake (T4, corresponding to the maximum concentration, C_max_). After centrifugation at 3500 rcf for 5 min, plasma was recovered from whole blood samples. Plasma samples were collected and stored at −80 °C until analysis.

According to the Kaftrio^®^ summary of product characteristics (SmPC) [22], patients received a morning dose consisting of two ivacaftor 75 mg/tezacaftor, 50 mg/elexacaftor, 100 mg tablets, whereas one ivacaftor 150 mg tablet was taken in the evening (every 12 h).

Informed consent was obtained from parents or legal guardians of patients aged < 18 years. This report was carried out in accordance with the Declaration of Hesinki within a TDM program performed as routine clinical practice at the Bambino Gesù Children’s Hospital. The protocol was approved by the Ethics Committee of Bambino Gesù Children’s Hospital (3243/2023) on 3 November 2023.

### 4.5. Determination of ELX/TEZ/IVA Plasma Levels by LC-MS/MS

Elexacaftor/tezacaftor/ivacaftor plasma levels were determined using high-performance liquid chromatography (HPLC) coupled to mass spectrometry (MS/MS). EDTA–whole blood samples were collected before (T0) and 4 h after (T4) morning drug intake. Analysis of Kaftrio^®^ plasma levels was performed at Division of Metabolic Diseases and Drug Biology of Bambino Gesù Children’s Hospital, IRCCS, in Rome (Italy). The liquid chromatography (LC) system consisted of an UHPLC Agilent 1290 Infinity II (Agilent Technologies, Deutschland GmbH, Waldbronn, Germany). Gradient chromatographic separation was performed in reverse phase mode with a Luna Omega 1.60 µm Polar C18 (100 × 2.1 mm) column (Phenomenex) maintained at 40 °C, with mobile phase A (0.1% formic acid (FA) in water) and mobile phase B (0.1% formic acid and 25% of 2-propanol in ACN). The flow rate of mobile phases was 0.4 mL/min. The duration of each chromatographic run was 10.50 min. The gradient conditions are reported in Table 6.

The volume injected was 1 μL. Detection of IVA, IVA-d9, TEZ, TEZ-d4, ELX and ELX-d3 was carried out using a 6470 Mass Spectrometry system (Agilent Technologies, Deutschland GmbH, Waldbronn, Germany) equipped with an ESI-JET-STREAM source operating in the positive ion (ESI+) mode. Mass spectrometric conditions were as follows: Gas Temperature 350 °C, Gas Flow 9 L/min, Sheath Gas Temperature 300 °C, Sheath Gas Flow 9 L/min, Capillary 4000 V, Nebulizer 35 psi. Analytes were detected in multiple reaction monitor (MRM) mode transitions are displayed in Table 7.

Optimizer software 10.1 (Agilent Technologies, Deutschland GmbH, Waldbronn, Germany) was used to generate and optimize MRM transitions and mass spectrometer parameters, while MassHunter 10.1 (Agilent Technologies, Deutschland GmbH, Waldbronn, Germany) software was used for quantitative analysis. Structures of the investigated compounds and their fragments (identified as quantifier and qualifier compounds) are displayed in Appendix A.

### 4.6. Sample Preparation

A 10 µL measure of IS working solution with IVA-d9, TEZ-d4 and ELX-d3 was added to 50 µL of plasma or CAL or QC or blank sample in darkened tubes. After vortexing for 5 s, 300 µL of pure ACN were added to each sample. Samples were mixed for 60 s and centrifuged at 13,000 rpm for 9 min at room temperature. Thereafter, 200 µL of supernatant were collected in amber glass vials and injected into the UHPLC-MS/MS system.

### 4.7. Bioanalytical Method Validation

This method has been validated according to ICH guideline M10 on bioanalytical method validation and study sample analysis. 25 July 2022 EMA/CHMP/ICH/172948/2019, Committee for Medicinal Products for Human Use. Available at: https://www.ema.europa.eu/en/ich-m10-bioanalytical-method-validation-scientific-guideline (accessed on 28 August 2023) [41].

#### 4.7.1. Accuracy and Precision

For each QC level, intra- and inter-day assay accuracy and precision were estimated from 10 independent runs over a period of months. Accuracy was evaluated as the mean % bias; precision was reported as the % coefficient of variation (CV) at low, medium, high QC and LLOQ levels. The accuracy at each concentration level should be within ±15% of the nominal concentration, except at the LLOQ, where it should be within ±20%. The precision (%CV) of the concentrations determined at each level should not exceed 15%, except at the LLOQ, where it should not exceed 20%.

#### 4.7.2. Selectivity and Specificity

The absence of interfering peaks in blank biological matrix samples spiked with and without an internal standard, considered a hallmark of selectivity and specificity for method validation.

The median response of these blank samples should be below 20% of the LLOQ and not more than 5% of the IS response in the LLOQ sample to ensure the selectivity of the method. Blank samples were made with drug-free plasma samples collected from healthy donors recruited at the Blood Transfusion Centre of Bambino Gesù Children’s Hospital after obtaining informed consent.

#### 4.7.3. Carry-Over

Carry-over was assessed by analyzing plasma blank samples in triplicate after the ULOQ. According to ICH guideline M10 on bioanalytical method validation and study sample analysis guidelines, carry-over is considered absent when it is not higher than 20% of the analyte response at the LLOQ and 5% of the response for the IS.

#### 4.7.4. Matrix Effect and Extraction Recovery

The matrix effect (ME) and extraction recovery (ER) were measured for IVA, TEZ and ELX at low, medium and high QC levels by analyzing six different pools of blank matrix (neat) from individual healthy donors. ME was determined as B/A × 100% where B is the peak area of each analyte spiked to a blank matrix extract (spiking post-extraction) and A is the peak area of the analyte prepared at the same concentration in a pure solution [61]. ER was measured as C/B × 100%, where C is the peak area of each analyte in a blank matrix (spiked before the extraction). Ranges between 85–115% and 90–110% were considered acceptable for ME% and ER%, respectively. ME% and ER% were also evaluated after normalization for deuterated internal standard (IS-normalized).

#### 4.7.5. Stability

Autosampler stability was tested by analyzing IVA, TEZ and ELX concentrations in QCs maintained at room temperature at time 0 (corresponding to preparation of QC samples and to the first measurement) and after 24 h (short-term stability) or 9 days (long-term stability). We also evaluated, on prepared low-, medium- and high-QC samples, the stability following one cycle of freezing (−80 °C) and thawing (after 12 days). To further assess long-term stability, n = 3 T0 and T4 of authentic samples, stored at −80 °C over a period of five months, were randomly chosen and re-analyzed. The percentage difference between the concentration measured at each sampling point and the initial concentration was calculated. Based on ICH guideline M10, a percentage difference below the 15% was considered acceptable.

### 4.8. Evaluation of Method Greenness, Blueness and Whiteness

The analytical greenness of our method was evaluated using the AGREE tool available at https://agree-index.anvil.app/ (accessed on 25 June 2025) [45], in which every criterion reflects a principle of the green analytical chemistry (SIGNIFICANCE). The width of each segment in the AGREE pictogram indicates the weight assigned (from 1 to 4) to each principle in accordance with the attributed relevance. Here, we have applied a weight of 2 to criteria 1, 3, 4, 5, 8, 9, 10, 11, and 12, whereas a weight of 3 was assigned to criteria 2, 6 and 7, in order to highlight the relevance of a small sample size, absence of derivatization step and reduced volume of generated analytical waste, respectively. The performance of the analytical procedure for each principle is reflected by the color in the segment with the number corresponding to each criterion [45]. The overall score is reported in the middle of the pictogram with values close to 1 and dark green color indicating the highest greenness.

The practicality of our developed method was assessed using the Applicability Grade Index (BAGI) [46]. This index was calculated online at the official website https://bagi-index.anvil.app (accessed on 24 June 2025). The BAGI tool generates two types of results: an asteroid pictogram (as a graphical representation) and a numerical rating score. The BAGI metric incorporates 10 criteria to assess the practicability and greenness of different assays. For methods considered as “practical,” the final rating score should exceed 60 [46].

Finally, in order to assess the whiteness of our method, we used the new and improved version of the RGB model, called RGBfast, which includes ChlorTox Scale as one of the criteria [48]. The Excel file used for whiteness calculation can be found at doi:10.1016/j.greeac.2024.100120 (accessed on 25 June 2025).

### 4.9. Statistical Analysis

Demographic data and PK parameters were analyzed using descriptive statistics. The median with interquartile range (IQR) was used to describe C_trough_ levels. For IVA, the area under the curve 0–12 h (AUC_0–12h_) was also calculated after estimating plasma concentration at 12 h as follows: (C12h): C4h*e^−β(t12−t4)^ [62]. Spearman’s r was used to evaluate the correlation between C_trough_ and AUC values. All statistical analyses were performed using GraphPad Prism v.10 (GraphPad Software, San Diego, CA, USA). Statistical significance was set at *p* < 0.05.

## 5. Conclusions

In conclusion, we believe that the bioanalytical method presented and validated here could be used for contemporary measurement of ivacaftor, tezacaftor and elexacaftor plasma levels and, therefore, it could be adopted for future PK/PD studies and for TDM application during the routine clinical practice. In this regard, in the *era* of precision medicine, our knowledge not only on the clinical efficacy but also on the tolerability and long-term safety profile of “caftors” could be significantly improved through the implementation of TDM procedures. Considering also the high cost burden of these medications to the health system, a TDM-based approach could efficiently meet the clinical needs of CF patients while at the same time facilitating most cost-effective management. Finally, based on the recent approval by regulatory authorities of ETI combination for treating CF patients aged 2 years and older carrying at least one F508del mutation, future studies will be required to collect additional TDM-real world data from younger CF patients.

## Figures and Tables

**Figure 1 pharmaceuticals-18-01028-f001:**
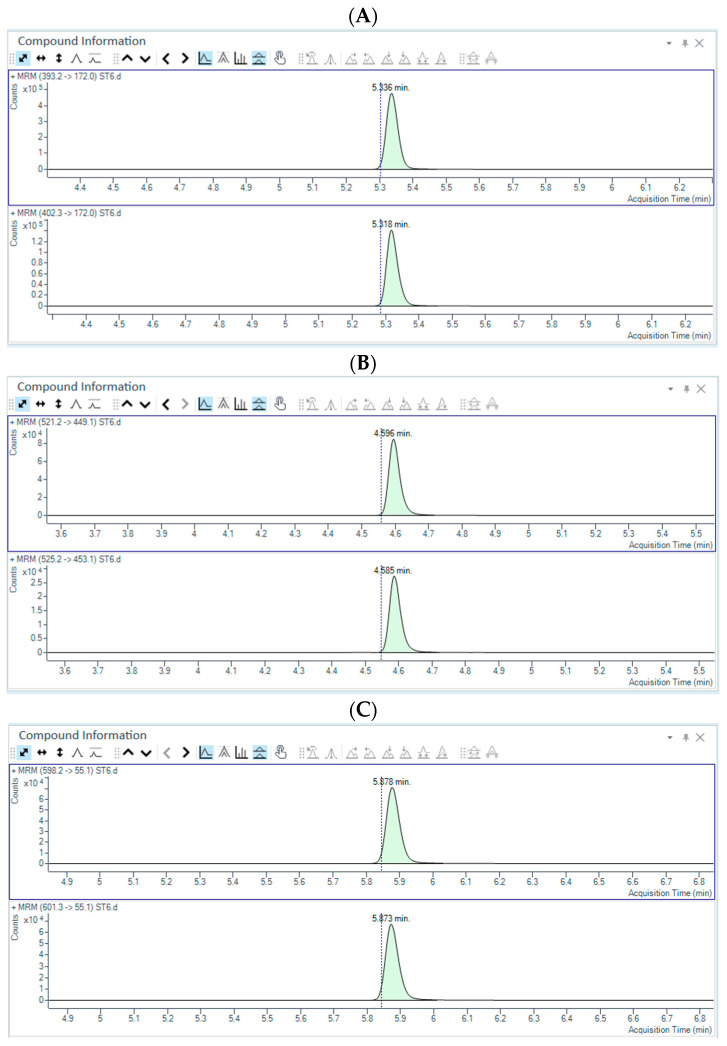
Chromatograms of plasma calibration point 6 (ST6) for ivacaftor (**A**), tezacaftor (**B**) and elexacaftor (**C**). For each chromatogram, the upper and lower layers indicate the fragments used as the quantifier and the internal standard compound, respectively. The relative response (counts) from the baseline and the acquisition time (min) are reported on the y- and x-axes, respectively. For each peak, the retention time is displayed.

**Figure 2 pharmaceuticals-18-01028-f002:**
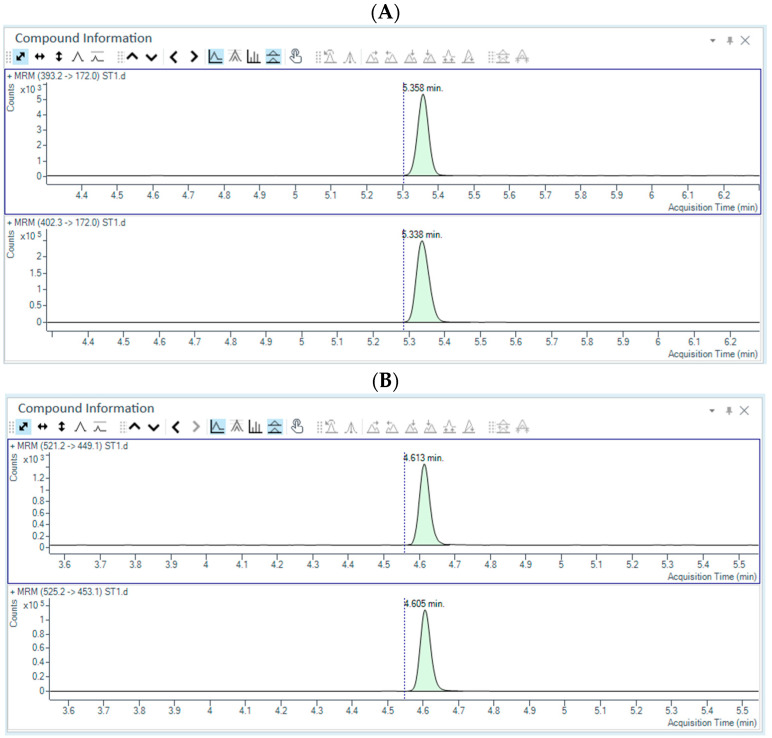
Chromatograms of LLOQ for ivacaftor (**A**), tezacaftor (**B**) and elexacaftor (**C**). For each chromatogram, the upper and lower layers indicate the fragments used as the quantifier and the internal standard compound, respectively. The relative response (counts) from the baseline and the acquisition time (min) are reported on the y- and x-axes, respectively. For each peak, the retention time is displayed.

**Figure 3 pharmaceuticals-18-01028-f003:**
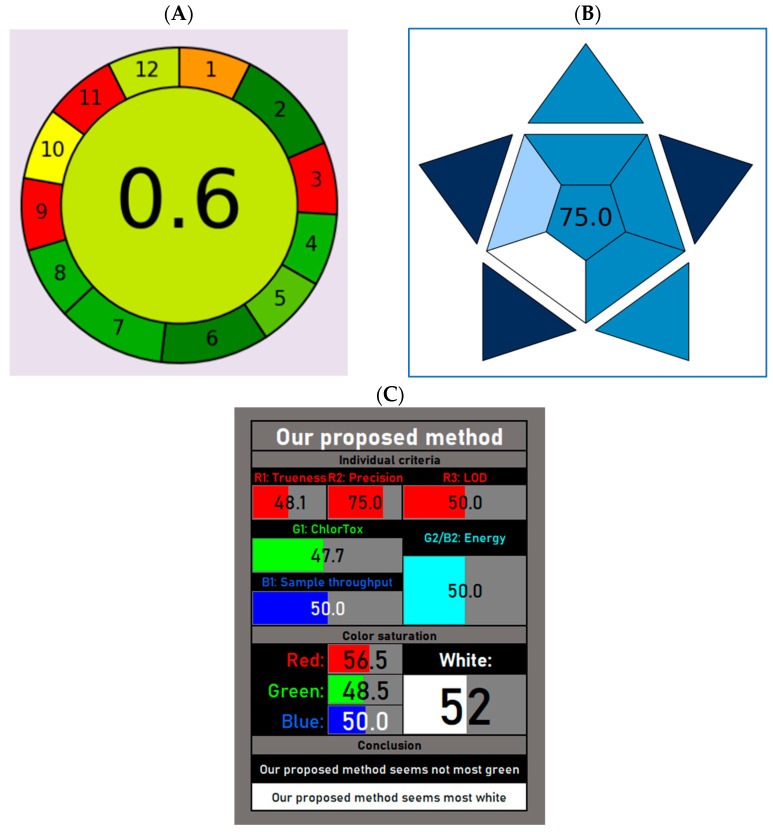
(**A**) Greenness profiles (AGREE score) of the proposed method. (**B**) Blueness assessment of our method using the BAGI tool. (**C**) Automatically formatted tables/pictograms obtained for our method using the RGBfast model.

**Figure 4 pharmaceuticals-18-01028-f004:**
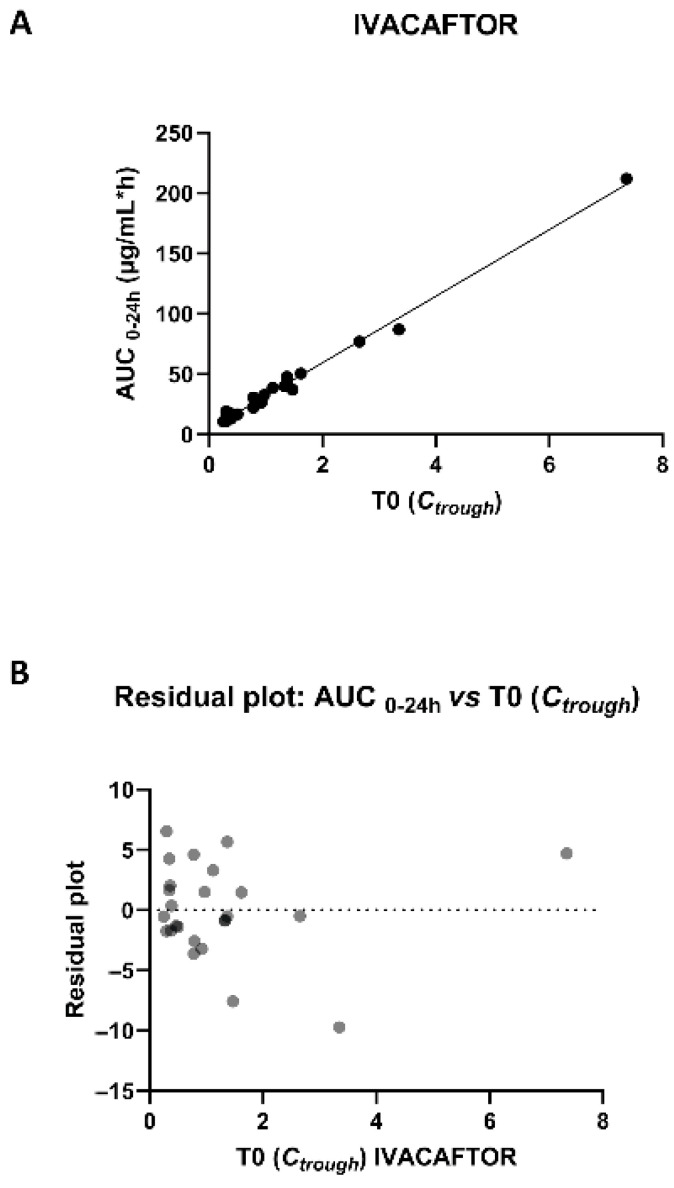
(**A**) AUC_0–24h_ versus C_trough_ for ivacaftor. Spearman r = 0.94 (95% confidence interval, 0.86–0.97), R^2 = 0.99, regression equation: y = 27.6 * x + 4.30. The dotted blue lines indicate 95% confidence intervals (CIs). (**B**) Residual plot: simple linear regression of AUC_0–24h_ versus C_trough_ for ivacaftor. The % CV between the measured and predicted AUC_0–24h_ was 0.02 ± 13.12 (mean ± SD).

**Table 1 pharmaceuticals-18-01028-t001:** Intra-assay accuracy and precision of IVACAFTOR, TEZACAFTOR and ELEXACAFTOR.

**IVACAFTOR**	**Parameter**				
	**Quality Control Sample (target concentration)**	**LLOQ (0.1 µg/mL)**	**L-QC (0.30 µg/mL)**	**M-QC (8.0 µg/mL)**	**H-QC (18.0 µg/mL)**
	**Number of analyzed samples**	10	10	10	10
	**Ivacaftor concentration found µg/mL (median, range)**	0.10 (0.10–0.10)	0.32 (0.31–0.33)	7.67 (7.61–7.86)	15.59 (15.38–15.69)
	**Intra-assay %bias**	2.00	6.66	−3.82	−13.61
	**Intra-assay %CV**	1.20	3.13	1.36	0.75
**TEZACAFTOR**	**Quality Control Sample (target concentration)**	**LLOQ (0.1 µg/mL)**	**L-QC (0.30 µg/mL)**	**M-QC (8.0 µg/mL)**	**H-QC (18.0 µg/mL)**
	**Number of analyzed samples**	10	10	10	10
	**Ivacaftor concentration found µg/mL (median, range)**	0.10 (0.10–0.10)	0.29 (0.28–0.29)	7.84 (7.74–7.89)	15.90 (15.76–15.99)
	**Intra-assay %bias**	−2.4	−4.0	−2.22	−11.73
	**Intra-assay %CV**	1.17	1.55	0.77	0.64
**ELEXACAFTOR**	**Quality Control Sample (target concentration)**	**LLOQ (0.1 µg/mL)**	**L-QC (0.30 µg/mL)**	**M-QC (8.0 µg/mL)**	**H-QC (18.0 µg/mL)**
	**Number of analyzed samples**	10	10	10	10
	**Ivacaftor concentration found µg/mL (median, range)**	0.10 (0.09–0.10)	0.29 (0.28–0.29)	7.91 (7.81–7.92)	15.71 (15.52–15.82)
	**Intra-assay %bias**	−2.4	−4.7	−1.57	−12.91
	**Intra-assay %CV**	4.44	1.92	0.68	0.81

**Table 2 pharmaceuticals-18-01028-t002:** Inter-assay accuracy and precision of IVACAFTOR, TEZACAFTOR and ELEXACAFTOR.

**IVACAFTOR**	**Parameter**				
	**Quality Control Sample (target concentration)**	**LLOQ (0.1 µg/mL)**	**L-QC (0.30 µg/mL)**	**M-QC (8.0 µg/mL)**	**H-QC (18.0 µg/mL)**
	**Number of analyzed samples**	10	10	10	10
	**Ivacaftor concentration found µg/mL (median, range)**	0.10 (0.10–0.11)	0.32 (0.31–0.32)	7.88 (7.16–8.18)	15.53 (15.37–15.62)
	**Inter-assays %bias**	3.00	6.00	−3.85	−13.82
	**Inter-assays %CV**	6.51	1.41	6.21	0.66
**TEZACAFTOR**	**Quality Control Sample (target concentration)**	**LLOQ (0.1 µg/mL)**	**L-QC (0.30 µg/mL)**	**M-QC (8.0 µg/mL)**	**H-QC (18.0 µg/mL)**
	**Number of analyzed samples**	10	10	10	10
	**Ivacaftor concentration found µg/mL (median, range)**	0.10 (0.09–0.10)	0.29 (0.28–0.29)	8.20 (7.36–8.34)	15.89 (15.61–16.01)
	**Inter-assays %bias**	−3.60	−4.67	−0.63	−11.98
	**Inter-assays %CV**	2.99	1.92	5.71	1.01
**ELEXACAFTOR**	**Quality Control Sample (target concentration)**	**LLOQ (0.1 µg/mL)**	**L-QC (0.30 µg/mL)**	**M-QC (8.0 µg/mL)**	**H-QC (18.0 µg/mL)**
	**Number of analyzed samples**	10	10	10	10
	**Ivacaftor concentration found µg/mL (median, range)**	0.10 (0.09–0.10)	0.28 (0.28–0.29)	8.41 (7.63–8.49)	15.76 (15.61–16.01)
	**Inter-assays %bias**	−3.20	−5.33	1.70	−12.36
	**Inter-assays %CV**	2.67	1.93	5.20	1.11

**Table 3 pharmaceuticals-18-01028-t003:** Results of matrix effect (ME) and extraction recovery (ER) experiments for IVACAFTOR, TEZACAFTOR and ELEXACAFTOR.

**IVACAFTOR**		**L-QC (0.3 µg/mL)**		**H-QC (18 µg/mL)**	
		**ER%**	**ME%**	**ER%**	**ME%**
		91	96.2	91	85.9
	**Number of analyzed samples**	3	3	3	3
**TEZACAFTOR**		**L-QC (0.3 µg/mL)**		**H-QC (18 µg/mL)**	
		**ER%**	**ME%**	**ER%**	**ME%**
		95	96.5	92.50	85.2
	**Number of analyzed samples**	3	3	3	3
**ELEXACAFTOR**		**L-QC (0.3 µg/mL)**		**H-QC (18 µg/mL)**	
		**ER%**	**ME%**	**ER%**	**ME%**
		92.6	95.10	94.3	96.41
	**Number of analyzed samples**	3	3	3	3

**Table 4 pharmaceuticals-18-01028-t004:** Short- and long-term autosampler stability for IVACAFTOR, TEZACAFTOR and ELEXACAFTOR.

**IVACAFTOR**	**Time Point**	**Time 0**	**24 H**	**Day 9**
	**Measured concentration for L-QC (0.3 µg/mL)**	0.330	0.335	0.250
	**Stability (%)**		101.51	75.75
	**Measured concentration for M-QC (8 µg/mL)**	8.56	8.55	6.70
	**Stability (%)**		99.88	78.27
	**Measured concentration for H-QC (18 µg/mL)**	15.50	15.70	11.40
	**Stability (%)**		101.29	73.54
**TEZACAFTOR**	**Time Point**	**Time 0**	**24 H**	**Day 9**
	**Measured concentration for L-QC (0.3 µg/mL)**	0.29	0.28	0.30
	**Stability (%)**		96.55	103.79
	**Measured concentration for M-QC (8 µg/mL)**	8.26	8.27	8.70
	**Stability (%)**		100.12	105.32
	**Measured concentration for H-QC (18 µg/mL)**	15.78	16.00	16.70
	**Stability (%)**		101.39	105.83
**ELEXACAFTOR**	**Time Point**	**Time 0**	**24 H**	**Day 9**
	**Measured concentration for L-QC (0.3 µg/mL)**	0.29	0.295	0.33
	**Stability (%)**		101.72	113.79
	**Measured concentration for M-QC (8 µg/mL)**	8.40	8.38	9.50
	**Stability (%)**		99.76	113.09
	**Measured concentration for H-QC (18 µg/mL)**	15.90	15.70	18.20
	**Stability (%)**		98.74	114.46

**Table 5 pharmaceuticals-18-01028-t005:** PK exposure parameters for IVA, TEZ and ELX.

**PK Parameter**	**IVA**		**TEZ**		**ELX**	
Number of Values	n = 25		n = 25		n = 25	
**Sampling Time Point**	**T0**	**T4**	**T0**	**T4**	**T0**	**T4**
**25th Percentile (µg/mL)**	0.37	1.07	0.72	3.75	5.82	13.27
**Median (µg/mL)**	0.79	1.49	1.37	4.52	8.73	17.85
**75th Percentile (µg/mL)**	1.34	2.43	1.87	6.32	12.94	24.10
**Measured AUC_0–12h_ (µg/mL*h)**						
**Number of Values**	n = 25		n.a		n.a	
**25th Percentile**	5.33		n.a		n.a	
**Median**	11.43		n.a		n.a	
**75th Percentile**	19.45		n.a		n.a	

n.a = not applicable (i.e., AUC0–24 h was not calculated for TEZ and ELX); T0 = blood samples collected before morning drug assumption (C_trough_); T4 = blood samples collected 4 h after the morning dose intake (corresponding to the time to reach the maximum plasma concentration, C*max*); AUC (area under the curve) calculated within the dose interval for IVA (12 h).

**Table 6 pharmaceuticals-18-01028-t006:** Gradient elution conditions.

Time (min)	Mobile Phase B (%)
0	30
0.5	30
4	85
7	85
7.5	30
10.5	30

**Table 7 pharmaceuticals-18-01028-t007:** MRM transitions for the analysed compounds.

Compound	MRM Transitions (*m*/*z*)
**IVA (quantifier)**	393.2 → 172.0
**IVA (qualifier)**	393.2 → 337.1
**IVA-d9 (quantifier)**	402.3 → 172.0
**IVA-d9 (qualifier)**	402.3 → 328.2
**TEZ (quantifier)**	521.2 → 449.1
**TEZ (qualifier)**	521.2 → 131.0
**TEZ-d4 (quantifier)**	525.2 →453.1
**TEZ-d4 (qualifier)**	525.2 →135.0
**ELX (quantifier)**	598.2 → 55.1
**ELX (qualifier)**	598.2 → 422.2
**ELX-d3 (quantifier)**	601.3 → 55.1
**ELX-d3 (qualifier)**	601.3 → 422.2

## Data Availability

The original contributions presented in this study are included in the article/Appendix A. Further inquiries can be directed to the corresponding authors.

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
