# Peer review of "Development and Validation of a New LC-MS/MS Method for Simultaneous Quantification of Ivacaftor, Tezacaftor and Elexacaftor Plasma Levels in Pediatric Cystic Fibrosis Patients"

_pharmaceuticals, 2025, doi:10.3390/ph18071028_

Round 1
Reviewer 1 Report
Comments and Suggestions for Authors
Comments:
This work is very interesting, well done and well written. I recommend publishing after caring for the few next comments please.
- Line 27: fell not felt.. please check language in whole paper once again.
- Intra-day and inter-days accuracy and precision were ≤15%: this is low precision performance, though this could be the case with biological samples, this could be quite serious if the therapeutic window for these drugs is low, hence it would be necessary to monitor blood levels accurately before deciding the next dose for patients. Please clarify in discussion.
- EMA guidelines does not include robustness testing, yet testing robustness is important when introducing new analytical procedure to the scientific community, where some analysis parameters could show some constraints in use, so if you can add it in validation section that will be better.
- Section 3.1. Linearity: there is a difference between linearity and calibration range, so please clearly indicate the two ranges separately, where ICH guidelines identify them as two different parameters. It is all about precision at borders of range, where calibration range guarantees that all its points offer precision in analysis, while this is not the case with linearity. Please adjust this section.
- There is no system suitability testing section provided in validation as well. Peak symmetry, capacity factor, resolution…etc; all of these parameters are quite important to be described when introducing new analytical methods. For example figure 1 (E), it could be noticed by mere eye that there is tailing in the peak, we need to express this in numerical values and check if it fits the acceptable limits, as tailing could mean there are some tiny impurity peaks hidden in this tail for example, hence affecting specificity of the method.. so please get back to system suitability testing parameters in USP and verify them in a separate table for each of your drugs.
Author Response
This work is very interesting, well done and well written. I recommend publishing after caring for the few next comments please.
Many thanks to Reviewer#1 for appreciating our work and for raising these interesting comments that we have addressed as following:
Line 27: fell not felt.. please check language in whole paper once again.
We do apologies for this typesetting mistake. Language has been revised throughout the whole manuscript.
Intra-day and inter-days accuracy and precision were ≤15%: this is low precision performance, though this could be the case with biological samples, this could be quite serious if the therapeutic window for these drugs is low, hence it would be necessary to monitor blood levels accurately before deciding the next dose for patients. Please clarify in discussion.
Many thanks to Reviewer#1 for raising this interesting point. According to The ICH guideline M10 on bioanalytical method validation and study sample analysis. 25 July 2022 EMA/CHMP/ICH/172948/2019, Committee for Medicinal Products for Human Use. Available at: https://www.ema.europa.eu/en/ich-m10-bioanalytical-method-validation-scientific-guideline, precision (defined as %CV) of the concentrations determined at each levels should not exceed 15%, except at the LLOQ, where it should not exceed 20%. In our manuscript, both intra-day and inter-days accuracy and precision for QC samples and LLOQ were within this limit. Perhaps, Reviewer#1 is referring to the therapeutic index of ivacaftor, tezacaftor and elexacaftor that is not mentioned in our manuscript and is out of the scope of our study. In fact, drugs with a narrow therapeutic index should be constantly monitored through TDM application.
EMA guidelines does not include robustness testing, yet testing robustness is important when introducing new analytical procedure to the scientific community, where some analysis parameters could show some constraints in use, so if you can add it in validation section that will be better.
Many thanks to Reviewer#1 for this precious suggestion. Since The ICH guideline M10 on bioanalytical method validation and study sample analysis does not include robustness testing, we have not assessed this validation parameter in our initial submission. However, in order to follow Reviewer#1 suggestion, we have evaluated robustness of our method by changing different variables as follows:
- Different Operators: Sample preparation and analytical run submission have been performed by
different operators in order to test the effects of inter-operator variability on drugs’ quantification among different analytical sessions;
- Room-Temperature Variations: Two analytical runs have been carried out by changing room temperature (25 ± 2 °C) to mimic changes that could partially affect instrument working conditions;
- Reagent Batch Changes: Mobile phases have been prepared by using reagents from different batches (including the acetonitrile that is used for both mobile phase B preparation and proteins’ precipitation);
- pH Variations in Mobile phase A: We performed two analytical runs by changing the pH of mobile phase A (2.70 ± 0.3) in order to evaluate the effect of pH variations on the measured drugs’ concentrations.
The results of these robustness tests indicated that our method remained reliable and produced consistent results under the tested conditions. A paragraph on the robustness testing has been included in the Bioanalytical method validation section of our revised manuscript.
Section 3.1. Linearity: there is a difference between linearity and calibration range, so please clearly indicate the two ranges separately, where ICH guidelines identify them as two different parameters. It is all about precision at borders of range, where calibration range guarantees that all its points offer precision in analysis, while this is not the case with linearity. Please adjust this section.
We do apologies for this misleading point. In the revised version of our manuscript we have modified the 2.1 paragraph (now entitled Calibration Curve and Linearity evaluation) in order to better elucidate the difference between linearity and calibration range.
There is no system suitability testing section provided in validation as well. Peak symmetry, capacity factor, resolution…etc; all of these parameters are quite important to be described when introducing new analytical methods. For example figure 1 (E), it could be noticed by mere eye that there is tailing in the peak, we need to express this in numerical values and check if it fits the acceptable limits, as tailing could mean there are some tiny impurity peaks hidden in this tail for example, hence affecting specificity of the method. so please get back to system suitability testing parameters in USP and verify them in a separate table for each of your drugs.
We’d like to thank Reviewer#1 for raising this point. As suggested, we have now included a supplementary table (Supplementary Table 1), in which system suitability parameters have been evaluated for each drug in accordance with USP (United States Pharmacopoeia, Chromatography, USP40-NF35, 2017). Results have been now mentioned in both Results and Discussion sections.
Reviewer 2 Report
Comments and Suggestions for Authors
The manuscript reports the development and validation of a new LC-MS/MS method for simultaneous quantification of ivacaftor, tezacaftor and elexacaftor plasma levels in pediatric cystic fibrosis patients. The importance of the investigated drugs is clearly and detailed discussed in the introduction also the necessity of TDM. Furthermore, the method validation is performed comprehensively and accurately according to the ICH guideline.
Major comments:
A lot of LC-MS or LC-MS/MS methods are described for CFTR modulators in literature. All of these methods were cited and their properties were summarized in a table to give a very good overview in comparison to the new one. In my opinion, it is also required to add the information whether a method is validated or not to the table. Additionally, the importance and necessity of the reported new method and its benefit should be more clearly highlighted in the discussion part of the manuscript.
In my opinion, a quadratic regression model should be tested for the calibrations. One possible option to evaluate the linearity of the calibrations is to analyze the residuals of the y-values. If linear regression describes the relationship of response and concentration best the residuals are normally distributed around zero. Could you please perform a residual analysis to assess the validity of the linear regressions for the calibration functions? And please add the R2 values to the text because it is very hard to read it in the insets of the diagrams.
From my point of view, it would be helpful to add the structures of the investigated compounds and the fragmentation to quantifier and qualifier ions to the manuscript.
Minor comments:
The name of the ICH guideline should be consistent all around the text. In the abstract the older name EMA guideline is used.
Why did you investigate the 9-day stability at room temperature? In my opinion, it is very improbably to perform a measurement over 9 days. At this point, it would be more interesting to investigate the stability over the duration of a normal batch (analytical run). It is important to know if the samples could be reanalyzed (without new sample preparation) if an analytical run has to be rejected.
Comments on the Quality of English LanguageI found a small number of errors all around the text (missing words or letters, commas ect.):
Line 21: name of the drug
Line: 46: "affecting upper a lower airway" - "and" instead of "a"?
line 51: missing comma after inflammation
line 115: "and" or "," before formic acid is missing
line 183: "were" instead of "was"
line 196: "injected" instead of "analyzed"
line 217: "were" before collected should be removed
line 497: "for" after 6h is missing
line 520: "in" should be removed
Author Response
Please, find attached the replies to Reviewer #2.

Reviewer 3 Report
Comments and Suggestions for Authors
This paper presents the development and validation of an LC-MS/MS method for the simultaneous quantification of ivacaftor (IVA), tezacaftor (TEZ), and elexacaftor (ELX) in pediatric cystic fibrosis (CF) patients. The study addresses an important gap in Therapeutic Drug Monitoring (TDM) for CFTR modulators, particularly in real-world clinical settings. Real applications are well appreciated. The method follows EMA validation guidelines, demonstrating reliability for PK/PD studies and clinincal monitoring. However, many revisions are required particularity to demonstrate novelty as many old LC-MS methods were reported for the same drugs.
- In the abstract, wrote full names for all abbreviations when first mentioning PK, PD, EMA
- Write more about novelty and merits for the new LC-MS/MS if compared to old ones in terms of methods and sensitivity.
- In the abstract, add numerical values for results subsection and more details for methods highlighting novelty over related articles particularly https://doi.org/10.3390/biomedicines11020628
Simultaneous Quantification of Ivacaftor, Tezacaftor, and Elexacaftor in Cystic Fibrosis Patients’ Plasma by a Novel LC–MS/MS Method.
- Date about linear range and LOQ should be included. Provide more detailed PK parameters (e.g., Cpmax, T max, AUC) in the results section.
- Detailed comparison with the aforementioned article in terms of method greenness ( using AGREE tool ), method bluness ( using BAGI ), method whiteness (RBG columns) is strongly recommended to highlight merits of the new method
- In sample preparation, the author used acetonitrile for protein precipitation while in old reported method methanol was used. This should be added for study limitation.
- In the abstract, correct typo error ivcaftor to be ivacaftor
- In the introduction, the authors should refer to this point “ Were there age-dependent differences in PK parameters (e.g., younger vs. older children)?
- In line 39-40, add Switzerland according to ref 1 study.
- In line 102, ICH guidelines while in the abstract , EMA guidelines which is correct ?
- At the end of the introduction, highlight drawbacks of old LC-MS/MS methods. So, development of the new one is appreciated.
- Figure 1 size should be expanded. I recommend removeing calibration curves to supplementary file and displaying chromatograms only with its proper size as fig 1 only.
- Figure 2 sized should be enlarged to be readable.
- Simple pharmacokinetics figures for the three drugs should be illustrated concentration against time
- Lines 348 to 366, are repeated in the introduction, remove them.
- In the discussion include new paragraph to explain stability fell down exclusively for L-, M- and H-QC of ivacaftor (73.54-78.27).
- Expand future plans.
Author Response
Please, find attached the response file to Reviewer#3.

Round 2
Reviewer 3 Report
Comments and Suggestions for Authors
The authors did all required recommendations. I appreciate their efforts and responses. The paper could be published in the current form. Thanks